# Effect of HIT Components on the Development of Breast Cancer Cells

**DOI:** 10.3390/life11080832

**Published:** 2021-08-13

**Authors:** Li-Yu Chen, Gurunath Apte, Annerose Lindenbauer, Marion Frant, Thi-Huong Nguyen

**Affiliations:** 1Institute for Bioprocessing and Analytical Measurement Techniques, 37308 Heiligenstadt, Germany; Li-Yu.Chen@iba-heiligenstadt.de (L.-Y.C.); Gurunath.Apte@iba-heiligenstadt.de (G.A.); Annerose.Lindenbauer@iba-heiligenstadt.de (A.L.); Marion.Frant@iba-heiligenstadt.de (M.F.); 2Department of Infection Biology, Leibniz Institute for Natural Product Research and Infection Biology, 07745 Jena, Germany; 3Institute of Nanotechnology (INT) and Karlsruhe Nano Micro Facility, Karlsruhe Institute of Technology, 76131 Karlsruhe, Germany; 4Faculty of Mathematics and Natural Sciences, Technische Universität Ilmenau, 98694 Ilmenau, Germany

**Keywords:** platelet factor 4, monoclonal antibodies, heparin, thrombocytopenia, breast cancer cells

## Abstract

Cancer cells circulating in blood vessels activate platelets, forming a cancer cell encircling platelet cloak which facilitates cancer metastasis. Heparin (H) is frequently used as an anticoagulant in cancer patients but up to 5% of patients have a side effect, heparin-induced thrombocytopenia (HIT) that can be life-threatening. HIT is developed due to a complex interaction among multiple components including heparin, platelet factor 4 (PF4), HIT antibodies, and platelets. However, available information regarding the effect of HIT components on cancers is limited. Here, we investigated the effect of these materials on the mechanical property of breast cancer cells using atomic force microscopy (AFM) while cell spreading was quantified by confocal laser scanning microscopy (CLSM), and cell proliferation rate was determined. Over time, we found a clear effect of each component on cell elasticity and cell spreading. In the absence of platelets, HIT antibodies inhibited cell proliferation but they promoted cell proliferation in the presence of platelets. Our results indicate that HIT complexes influenced the development of breast cancer cells.

## 1. Introduction

Cancer incidence is rapidly growing worldwide and causes a vast amount of deaths. The leading reason for the high mortality of patients with cancer is metastasis which mainly comprises invasion, intravasation, and extravasation. The spread of cancer cells to tissues and organs leading to the formation of new tumors is the major reason concerning the death of most cancer patients. In the metastasis process, tumor cells penetrate the surrounding tissues and pass through the basement membrane and extracellular matrix. The blood vessels within the tumor’s vicinity provide a route for the detached cells to enter the circulatory system and metastasize to distant sites [1,2]. In the blood circulation, if these cells meet favorable conditions, they will adhere to a new location, initialize angiogenesis and proliferate to produce the secondary tumor [1,3].

Circulating tumor cells in the bloodstream reach other distant organs and grow, resulting in tissue dysfunction. However, only a low amount of cancer cells survive in circulation due to cell degradation, high shear forces [4], and immune surveillance of natural killer cells [5,6]. Unfortunately, the circulating cancer cells secrete multiple factors to activate the surrounding platelets, forming their stable protective cloak that enables their escape and migration in the blood vessels [7,8]. This unexpected alliance between cancer cells and platelets can induce hematogenous metastatic dissemination and facilitate thrombosis formation [9,10]. Gay et al. reported that platelets support tumor metastasis [11]. Activation of platelets and the coagulation system have a crucial role in the metastasis of cancer as activated platelets protect tumor cells from immune elimination and induce their arrest at the endothelium, hence, facilitating the establishment of secondary lesions [12].

Heparin is frequently used as an anticoagulant in cancer patients [13]. It has been shown that heparins regulate blood vessel growth and regression [14]. Multiple mechanisms involved in this regulation including interactions of heparins with peptide growth factors such as the fibroblast, VEGF, and FGFs as well as for angiogenic inhibitors such as thrombospondin and platelet factor 4 (PF4). These interactions stabilize growth factors [14]. However, up to 5% of patients receiving this drug have a side effect, the so-called heparin-induced thrombocytopenia (HIT) [15,16]. HIT develops due to a complex interaction among multiple factors such as heparin (H), PF4, platelet-activating anti-PF4/H antibodies (aPF4/H Abs), and platelets. As ultra-large but undesired PF4/H complexes can be formed between the positively charged chemokine PF4 and negatively charged heparin [17,18], the immune system develops aPF4/H-Abs against these complexes. The binding of PF4 to heparin results in its conformational change [19] and exposes the binding site(s) for aPF4/H Abs. Among these antibodies, some aPF4/H Abs are dangerous as they activate platelets and the clotting system, resulting in life-threatening thrombotic complications [16,17]. They are The HIT antibodies (HIT Abs). It is reported that in addition to patients receiving heparin, HIT Abs are also developed in some patients after major surgery without heparin therapy due to mechanical compression devices that are used for thrombosis prophylaxis [20,21]. Recently, HIT Abs were frequently detected in severe COVID-19 infected patients [22,23] as well as in individuals vaccinated with several types of COVID-19 vaccine (AstraZeneca and Johnson & Johnson) [24,25]. The Fab parts of HIT Abs bind to PF4/H complexes bound on the platelet membrane while its Fc part links to a FcγRIIa receptor on another platelet [26,27]. Cross-linking of multiple HIT Abs-FcγRIIa pairs results in platelet aggregation and activation. With this mechanism, other healthy platelets are rapidly recruited into the prothrombotic process, leading to activation of the clotting cascade that releases thrombin and increases the risk for new thrombosis [28].

Two types of monoclonal antibodies (the KKO and RTO) that mimic the biological activity of human aPF4/H Abs are recently available [29]. KKO causes HIT in an animal model *in vivo* while RTO does not [30,31]. These antibodies have been used as models for studies of HIT to understand the binding characteristics of an antibody recognizing PF4/H complexes and activating platelets [32,33].

Recently, characteristics of aPF4/H Abs and other components in HIT have been thoroughly understood [34,35]. However, little is known about the role of the entire HIT complexes in cancer. Individually, it has been shown that binding of PF4 inhibits both FGF2 signaling and platelet activation [36], suppresses tumor growth and metastasis, and also induces apoptosis in myeloma cells [37]. The reduction of platelet numbers and platelet activation is known to inhibit tumor metastasis in mice [38,39]. The formation of platelet protective cloak for cancer cells has been clearly described, however, the molecular mechanism of cancer spread remains complex and challenging [40].

In addition to the basic side effect HIT that occurs in up to 5% of patients with heparin therapy. Cancer patients and cancer patients with COVID-19 infection, who receive heparin treatment or after COVID-19 vaccination [24,25] have a high risk of developing HIT and autoimmune HIT antibodies [22,23]. However, it has not yet been revealed whether or not the entire HIT complexes play a role in cancer development. We hypothesize that HIT complexes can adhere to breast cancer cells and bridge platelets that affect the mechanical properties of cells and influence cell growth. Interaction between the tumor cells and the surrounding stroma is extremely important in the development of tumor angiogenesis [6]. Micrometastases that may compose of HIT components develop in many patients but those are hard to detect by conventional techniques. This limitation makes metastasis the most life-threatening event in patients with cancer [3].

Here, we investigated the effect of HIT components including heparin, PF4, PF4/H complexes, and the combination of PF4/H complexes with RTO, KKO antibodies on the development of breast cancer cells. The change in mechanical properties of cells was identified via evaluation of cell elasticity over time utilizing nanoindentation-based atomic force microscopy (AFM) whereas protein-induced morphological changes in cells were quantified with confocal laser scanning microscopy (CLSM) imaging. Cell proliferation was analyzed to correlate the alterations in elasticity with the morphology of cells. We found that PF4 and KKO caused the strongest effect as they stiffened cells, reduced cell spread, and inhibited cell proliferation. In the presence of platelets, HIT antibodies together with PF4/H complexes enhanced proliferation. Our results indicate that HIT components can enhance thrombotic thrombocytopenia complications that may promote metastasis in cancer-associated HIT patients.

## 2. Material and Methods

### 2.1. Cell Culture

The human breast cancer cell line MDA-MB-231 (Leibniz Institute, DSMZ-German Collection of Microorganisms and Cell Cultures GmbH, Braunschweig, Germany) was trypsinized by 0.25% Trypsin-0.02% EDTA (PAA-BioPharm, Darmstadt, Germany) in PBS for 2 min at 37 °C followed by rinsing with PBS (PAA-BioPharm, Darmstadt, Germany). A concentration of 5000 cells was cultured on sterile 35 mm round glass Petri discs (Life Technologies, Darmstadt, Germany) in Dulbecco’s modified Eagle’s medium (DMEM) (Pan-Biotech) supplemented with 1% Penicillin/Streptomycin, 1% L-glutamine, and 10% heat-inactivated fetal calf serum (PAA-BioPharm, Darmstadt, Germany) at 37 °C under a 5% CO_2_ humidified atmosphere for 1–2 days before experiments. Cells were then coated with PF4 (Chromatech, Greifswald, Germany) of 20 µg/mL or PF4/H complexes pre-formed by 20 µg/mL PF4 with 0.5 U/mL unfractionated heparin (Heparin-Natrium-25000, Ratiopharm GmbH, Ulm, Germany), or 10 µg/mL aPF4/H Abs including RTO and KKO (BIOZOL, München, Germany) in culture medium for 1 h, 37 °C before AFM-indentation measurements. The measurements were carried out on the named days from the simultaneously started parallels.

### 2.2. AFM Nanoindentation

Nanoindentation measurements on breast cancer cell passages between 12 and 13 were carried out in a cell culture medium at 37 °C by AFM Nano Wizard 4 (JPK Instruments AG, Berlin, Germany). An inverted microscope (Olympus IX 81, objective 20 × 0.45) was utilized to position the cantilever tip on the cell surface. A commercial gold-coated silicon bead attached at the end of a cantilever (CP-CONT-AU-A, Nanoandmore GmbH, Germany) with a bead diameter of 2.62 ± 0.04 µm was exposed for 30 min in a UV-Ozone cleaner to create a hydrophilic surface. The bead was then coated with 1 mg/ mL silane-polyethylene glycol hydroxy linker (Silane-PEG-OH, PEG Mw 3400 Da, Nanocs, Boston, MA, USA). The nominal spring constant of the cantilevers of 0.02–0.77 N/m was calibrated on day 1 right before the start of the measurement using the thermal noise method (JPK Instruments AG), showing a spring constant of 0.321–0.55 N/m. Force curves were generated with a speed of 2 µm/s and a loading force of 1000 pN. For each condition, up to 500 force curves were recorded on top (nuclear region) of 14 single cells. Force curves were then converted to force vs. indentation curves, which were the slope difference of the F-D curves measured on top of the cell and a glass surface, followed by fitting with Hertz model integrated with JPK analysis software version spm-4.4.18+. This model describes a dependency of cell elasticity modulus (E) on indentation force (F) using Equation (1):(1)$$F=\frac{4E\sqrt{R}}{3\left(1-\rho^{2}\right)}\delta^{\frac{3}{2}}$$
where R, δ, and ρ are the radius of the bead, indentation depth, and Poisson’s ratio (ρ = 0.5 for a completely elastic soft sample), respectively. Data analyses were performed with origin software version 9.1 (Origin Lab Corporation, Northampton, MA, USA).

### 2.3. Cell Morphological Change Imaged by CLSM

The sample of 500 cancer cells was seeded on 96 wells plate and incubated with PF4/H complexes performed by PF4 (20 µg/mL final) (Chromatec, Greifswald, Germany); with unfractionated heparin (Heparin-Natrium-25000, Ratiopharm, Ulm, Germany) (0.5 U/mL final), and RTO or KKO antibody (10 µg/mL final) for 3 and 6 days, 37 °C. Cells were firstly washed twice with Dulbecco’s phosphate-buffered saline (DPBS) (PAN- Biotech GmbH) and 4% of paraformaldehyde (PFA) was added for 20 min to fix the cells. PFA was removed and incubated with permeable buffer for 10 min. Phalloidin DY590 (Mobitec GmbH, Göttingen, Germany) (1:20 dilution) was added to the cells for 45 min incubation at RT in dark. Cell nuclei were stained by Hoechst 33258 (Thermo Fisher, Germany) (1 µg/mL for 15 min in the dark at RT. Results were taken by confocal laser scanning microscope Zeiss LSM 710 (Carl Zeiss, Göttingen, Germany) and the fluorescence signal was quantified by ImageJ.

### 2.4. Isolation of Human Platelets

Human blood from healthy donors who were drug-free within the previous 10 days was collected into a tube of ACD-A 1.5 mL BD-Vacutainer (Fresenius Kabi, Bad Homburg, Germany) as previously described [41]. The blood tube was sealed with parafilm and rested at room temperature for 15 min. Platelet-rich plasma (PRP) was first obtained by centrifugation at 120× *g* for 20 min at room temperature. To isolate platelets, PRP in the presence of 15% acid-citrate dextrose (ACD-A, Fresenius Kabi, Germany) and 2.5 U/mL Apyrase (grade IV SIGMA, Munich, Germany) was centrifuged at 650× *g* for 7 min. The platelet pellet was resuspended in buffer pH 6.3 composed of 137 mM NaCl, 2.7 mM KCl, 11.9 mM NaHCO_3_, 0.4 mM Na_2_HPO_4_, 2.5 U/mL Hirudin, and incubated 15 min, 37 °C before centrifuging at 650× *g* for 7 min. Platelet pellets were again carefully resuspended in suspension buffer and adjusted to a concentration of 300 × 10^9^/L using a blood counter (pocH-100i, SYSMEX, Möhnesee, Germany) and then rested for 45 min, 37 °C before use.

### 2.5. Cell Proliferation

A total of 1500 cells together with different HIT components were seeded in 12 wells or 24-well plates (Th. Geyer Hamburg GmbH and Co. KG, Hamburg, Germany) for 3 and 6 days and counted for proliferation rate. The effect of platelets on cell proliferation rate was prepared in the same process as mentioned above. After incubation of cells with HIT components for 1 h, 30 µL platelet samples (300,000/µL) were added to the cells and cultured for 3 and 6 days for proliferation rate measurement. To count the cells, the samples were rinsed twice with PBS. Then, 600 µL Trypsin/EDTA (Sigma T-4049, Darmstadt, Germany) was added to detach the cells. To stop this detachment reaction, 400 µL medium composed of Dulbecco’s modified Eagle’s medium, DMEM Sigma D-6546 + 1% penicillin/streptomycin, Sigma P-0781 + 1% L-glutamine, Sigma G-8540, and 10% fetal bovine serum (Thermo Fisher Scientific, Darmstadt, Germany) was added to the wells. After several times resuspension, cells were mixed with Trypan blue (Sigma T-8154, Darmstadt, Germany) for counting using either a counting chamber for the microscope or an electronic cell counter with at least 4 repetitions.

## 3. Results

### 3.1. Experimental Setup for Determination of Cell Elasticity by AFM Nanoindentation

To examine the mechanical properties of cells, we used a spherical gold-coated tip mounted at the end of a tipless cantilever and brought it on cells cultured on the glass Petri dish (Figure 1A). Figure 1B shows a principle for a force nanoindentation measurement on a glass surface (black) and a human breast cancer cell (red). Applying a force F to the cantilever, the bead indented the cell resulting in cantilever bending.

The bending of the cantilever is converted to force while the piezo movement is recorded as separation distance, (providing force–distance curves (Figure 1B). The difference in the slope of the force curves measured on the hard glass (black, Figure 1B) and soft cell (red, Figure 1B) surfaces shows a deformation depth (σ) of the cell. These force–indentation curves were fitted with the Hertz model to calculate Young’s modulus of the cell. To track the effect of HIT complexes on cell elasticity, we coated cells with different types of HIT components including heparin, PF4, PF4/H complexes, and with the entire HIT complexes composing of PF4/H complexes and aPF4/H Abs. Antibodies of different strengths such as RTO and KKO were added separately to cells precoated with PF4/H complexes (Figure 1C).

### 3.2. Effect of HIT Components on the Elasticity of Breast Cancer Cells

After 24 h incubation with HIT components, AFM nanoindentation experiments were performed on these cells. The same probe was utilized to indent cells coated with HIT components and two repetitions were performed. In total, up to 500 force curves were recorded on 14 cells. Among experiments, we observed the same trend of changes in cell Young’s modulus. However, the absolute values varied as different cell passages were used. This resulted in large standard deviations. Analysis of these force curves allowed us to determine the distribution of cell elasticity vs. frequency of indentation (Figure 2A). For each condition, data from both experiments were collected in a histogram distribution and fitted with the Gaussian model to determine the average value and corresponding standard error of cell elastic modulus (E) (Figure 2A). By comparing these values, we observed that HIT components modified cell elasticity depending on the type of protein-coated on cells (Figure 2B). After 24 h, heparin induced the weakest increase in E modulus, followed by PF4/H complexes and the combination of PF4/H/RTO but the highest value for PF4 and PF4/H/KKO complexes. 

We further investigated the effect of HIT components on the mechanical properties of cells overtime on day 3, and on day 6, and compared them with cell response on day 1. HIT components caused a change in the E modulus of the cells over time (Figure 3). For non-coated cells, a significant increase in the E modulus overtime was observed within 6 days of investigation (Figure 3A). E modulus increased regularly over time in the presence of heparin (Figure 3B). PF4 affected cells differently as it stiffened the cells on day 1 and then slightly reduced the cell’s E modulus on days 3 and 6 (Figure 3C). On days 1 and 6, PF4/H complexes increased the cell’s E modulus as compared to non-coated cells but cells were strongly stiffened on day 3 (Figure 3D). The RTO/PF4/H complexes overall induced an increase in the cell’s E modulus and show a similar pattern as PF4/H complexes, indicating no significant effect of RTO on PF4/H complexes coated cells (Figure 3E). KKO/PF4/H complexes strongly increased the cell’s E modulus on day 1 and intensively reduced E modulus on day 3, but then increased on day 6, indicating a clear effect of KKO to cells via binding to PF4/H complexes.

### 3.3. HIT Antibodies Enhance the Spreading of Cancer Cells

The above AFM nanoindentation results clearly showed that HIT components modified the mechanical properties of breast cancer cells. We next tested if the proteins have an impact on cell spreading. For this, cells were cultured in the presence of PF4, or PF4/H complexes, or a mixture of both PF4/H complexes and antibodies for 3 and 6 days (Figure 4A). These proteins induced slight changes in the size of cells as visualized by CLSM. Quantification of cell areas shows an effect of HIT components on cells (Figure 4B). Statistic comparison with cell control on day 3 showed a significant reduction in cell size caused by PF4 and PF4/H complexes but no difference induced by RTO and KKO in complexed with PF4/H complexes (Figure 4B, gray). On day 6, all samples showed an increase in cell size with respect to day 3 but no significant difference was observed among individual proteins added.

### 3.4. HIT Components together with Platelets Enhanced Cell Proliferation

The above experiments clearly showed that HIT components modified cell elasticity and morphology. We next tested if these modifications interfere with cell proliferation. The cells were cultured with the addition of HIT components for 3 and 6 days in the presence and absence of platelets and then stained with trypan blue to identify live from dead cells (Figure 5).

Without platelets, PF4, PF4/H complexes, and a combination of RTO with PF4/H complexes enhanced cell proliferation on day 3 whereas the combination of KKO with PF4/H complexes inhibited cell proliferation (Figure 5A, gray). On day 6, PF4, and KKO/PF4/H significantly inhibited cell growth as compared to untreated cells but less effect was observed with PF4/H treated cells. RTO/PF4/H showed a significant increase in cell proliferation but slightly lower than day 3 (Figure 5A, black). The PF4 and KKO on one hand modified cell stiffness and on the other hand inhibited cell proliferation, indicating a clear effect of these proteins on the development of cells. However, when the platelets were introduced to the samples of HIT components treated cells, cells enhanced proliferation on both days. Platelets alone also caused cell proliferation. On day 3, only PF4 and KKO/PF4/H inhibited slightly whereas other proteins promoted cell proliferation. On day 6, PF4 promoted cell proliferation, while PF4/H complexes and RTO or KKO in complexed with PF4/H slightly reduced proliferation (Figure 5B).

## 4. Discussion

Our results indicate that HIT components, especially the PF4 or KKO in complexed with PF4/Heparin, modified the mechanical property of breast cancer cells and interfered with cell spreading and proliferation. In the absence of platelets, PF4 alone and the pathogenic HIT antibody (KKO) showed a lower proliferation rate than the non-pathogenic HIT antibody (RTO). However, in the presence of platelets, the investigated proteins promoted cell proliferation. 

We started with the determination of changes in the elasticity of cells and found that the cells responded differently to HIT components. The results after day 1 of incubation with proteins, the E modulus values of cells other than those incubated with heparin increased as compared with non-treated cells. Heparin showed a linear increase in the cell E modulus when the incubation time was extended. This can be attributed to the changes in the cytoskeletal structure of the cells and may have been the result of the depletion of actin filaments [42]. To date, for the treatment of cancer patients with thrombosis, heparin is the classic anticoagulant of choice. It is also reported that patients with heparin administration improve cancer survival rates independently of the effect on thromboembolism [43]. With the ability to reduce cell–cell interaction, the most important effect of heparin is to prevent the metastatic cascade, which is responsible for most cancer-related deaths. Moreover, it inhibits tumor growth and angiogenesis as it can bind to different proteins such as growth factors and cytokines like FGF, VEGF, PF4, IL8, and adhesion proteins such as vitronectin, fibronectin, selectins, and integrins [43]. In addition, the ability of heparin in inhibiting the formation of stress fibers also interferes with the adherence of cancer cells to the endothelium [44,45]. It has been reported that a high concentration of heparin can disrupt the binding of PF4 in complexes with polyanions on the cell surface [34,46,47], indicating that heparin can inhibit cell proliferation caused by PF4 in the presence of platelets as we observed in Figure 5B.

Cells incubated with PF4 are softer along the longer incubation times. PF4 has been found to downregulate the vascular endothelial growth factor VEGF in breast cancer (TA3) and ovarian cancer (SKOV-3 and ES-2) cell lines, and thus, inhibit tumor growth and induce apoptosis [48]. The binding of PF4 to breast cancer cells may be charge-related as cancer cells exhibit a high density of polyanions such as chondroitin sulfate, polyphosphates, or glycosaminoglycan (GAG) [36]. It has been shown that cancer cells exhibit a highly negatively charged surface not only due to the presence of polyanions but also because of their metabolic characteristics of higher glycolysis rate which eventually results in the generation of certain enzyme cofactors [49]. However, the PF4 induces the strongest increase in E modulus only on day 1. The compelling effect of the cationic PF4 at the initial time point can be attributed to the fact that it is strongly attracted to the negatively charged cell membrane. The changes in cell stiffness correlate with cell spreading for PF4. In the absence of platelets, the PF4 caused a reduction in the size of cells and partially inhibited cell proliferation on day 3, and further enhanced the effect on day 6 (Figure 5A). Our results are in line with the previous study in the absence of platelets, showing that PF4 suppresses tumor growth and metastasis [37]. However, in the presence of platelets, PF4 promoted cell proliferation on day 6, indicating that PF4 can promote cancer metastasis if cells enter the blood circulation.

The ultra-large complexes formed between PF4 and heparin affected the cells differently as compared to the influence of PF4 or heparin alone. They stiffened cells on day 1 and enhanced the effect on day 3 before getting softer by day 6. This strong variation of stiffness was likely due to the existence of a distinct binding mechanism of the complexes formed between PF4 and heparin. Previous studies have proven that heparin forms a complex with PF4 and induces a conformational change in PF4, and therefore, exposes binding epitopes for binding of anti-PF4/H antibodies [19,50]. As our results show a different response of cells to PF4/H complexes than PF4 or heparin alone, we suspect that the conformational change in PF4 induced by heparin and large PF4/H complexes [51,52] caused a distinct binding mechanism of PF4/H complexes to the cell membranes. On day 3, these complexes also caused a reduction of cell spreading but facilitated cell proliferation in both the absence and presence of platelets, which is distinct from the effect of PF4 or heparin alone. Perhaps the large complexes can inhibit cell fibers but their interactions support cells to grow. 

When RTO antibodies were added to cells precoated with PF4/H complexes, a similar pattern of variation in cell stiffness as compared with the effect of PF4/H complexes was observed (Figure 3D,E), indicating no significant effect of RTO on cells. In contrast, KKO/PF4/H complexes strongly stiffened the cells on day 1 and soften the cell on day 3. It is likely that KKO stabilized PF4/H complexes and caused an additional effect on cells. Even though both RTO and KKO antibodies did not show a significant difference in inhibiting cell spreading (Figure 4), the RTO promoted cell proliferation whereas the KKO strongly inhibited cell proliferation on both day 3 and day 6 (Figure 5A). However, when platelets were added, both RTO and KKO induced cell proliferation (Figure 5B).

Even though the binding mechanism of RTO and KKO antibodies to breast cancer cells remains unclear, the RTO is known to bind only to a single PF4 monomer while the KKO is intercalated in between two PF4 monomers in a PF4/H complex [53]. The RTO has a lower affinity and weaker binding force to PF4/H complexes than the KKO [32,54]. These different binding characteristics of antibodies seem to correlate with the change in cell stiffness and resulted in a dissimilar effect on cancer cells. As KKO activates platelets via FcγRIIa while RTO does not [53], it also induces changes in cell elasticity differently. It is likely that initially, the KKO/PF4/H complexes contacted the cell membrane, and after that, they underwent another mechanism that softened cells and inhibited cell proliferation in the absence of platelets.

Our results indicate that HIT components including PF4, PF4/H complexes, and anti-PF4/Heparin antibodies (RTO, KKO), facilitate cell proliferation in the presence of platelets. Probably, the investigated proteins mediate between platelets and cells while bound platelets secrete growth factors for the cells to develop. However, only PF4 and KKO are known to allow binding platelets while RTO does not. Yet, RTO also slightly enhanced cell proliferation, meaning that only weak binding of this antibody to platelet is already sufficient cooperation with platelets to trigger the release of growth factor for cells [11,12].

Overall, HIT components-stiffen breast cancer cells and can facilitate cell penetration or invasion. Furthermore, cancer cells secrete multiple factors to activate the surrounding platelets, forming their stable protective cloak that enables their migration and escapes in the blood vessels [7,8]. This process can induce hematogenous metastatic dissemination and facilitate thrombosis formation [9,10,11,12]. Our findings indicated that cancer patients associated with HIT may suffer from higher risk and complex thrombosis formation as well as enhanced risk for cancer metastasis.

## 5. Conclusions

HIT components bind and modify the mechanical property of breast cancer cells that affect the development of cells. In the absence of platelets, only PF4 and HIT antibodies (the KKO) inhibited cell spreading and proliferation. However, in the presence of platelets, all HIT components promoted breast cancer cells to grow. Our findings indicate that HIT components mediate cancer cells, platelets, and potentially other blood cells that enhance the risk of thrombotic thrombocytopenia and facilitate cancer cell metastasis in cancer patients associated with HIT. A similar mechanism of HIT components might also be relevant for other types of cancers.

## Figures and Tables

**Figure 1 life-11-00832-f001:**
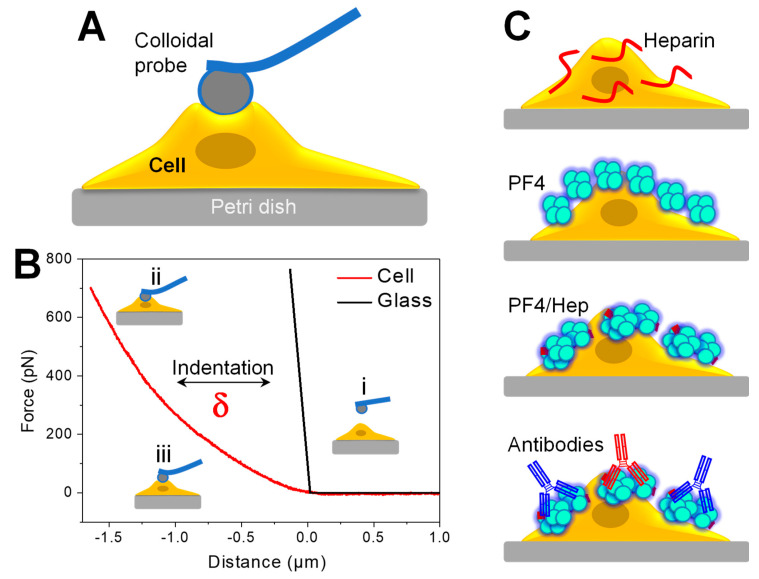
Experimental setup for measuring cell elasticity under the effect of HIT complexes by nanoindentation-based AFM. (**A**) Cartoon showing colloidal probe indents on a breast cancer cell cultured on a Petri dish. (**B**) Schematic showing approach force versus indentation curves measured on hard glass (black) and cell (red) showing (i) no interaction when the bead is far from the sample surface. The degree of cantilever bending is (ii) low or (iii) high depending on the distance that the piezo travels. Indentation depth (δ) is subtracted from the slopes of the force curves measured on the cell and glass surface. (**C**) Cartoons show cells coated with various HIT components before carrying AFM force indentation measurements.

**Figure 2 life-11-00832-f002:**
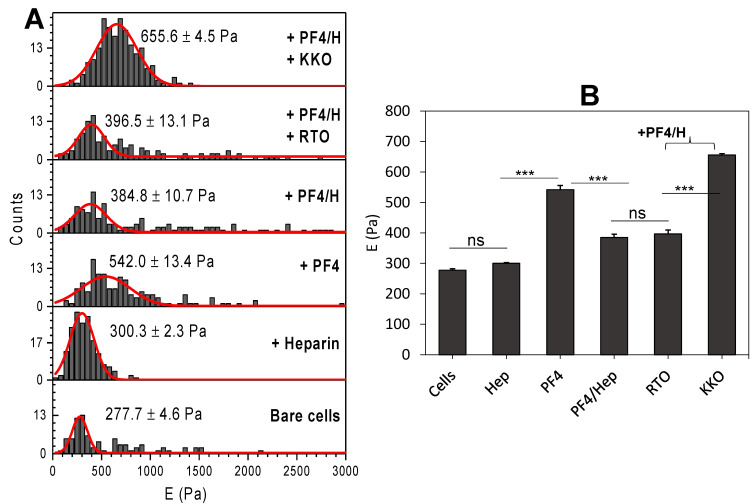
Elastic (E) modulus of breast cancer cells after 24 h incubation with HIT components. (**A**) Representative histograms of a single experiment show the elasticity of the cells changed after coating with different proteins. Gaussian fits (red) allow determination of averages ± standard errors at each condition which (**B**) were collected for comparison. Statistics by ANOVA tests obtained by comparing whole data points in the representative experiment: ns = no significant difference (*p* < 0.05), *** = significant difference (*p* < 0.05).

**Figure 3 life-11-00832-f003:**
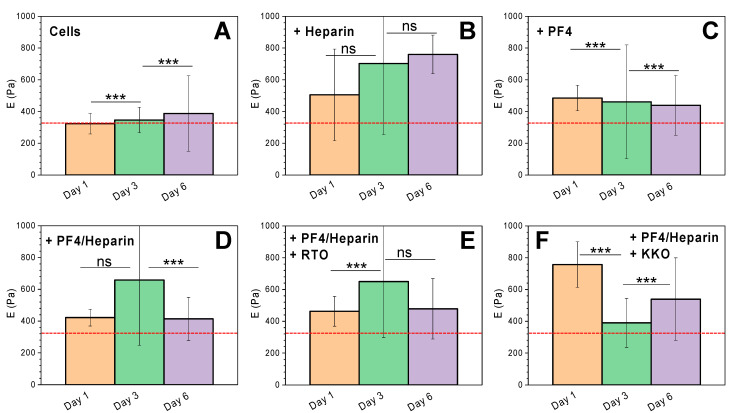
Comparison of **E** modulus of cells in the presence of each HIT component at different time points. (**A**) An increase in the E modulus of noncoated cells was observed as guided by the red dashed line. (**B**) E modulus increased over time in the presence of heparin. (**C**) PF4 stiffened the cells on day 1 and then slightly softened cells on days 3 and 6. (**D**) PF4/H complexes enhanced cell stiffness and a similar pattern was observed with (**E**) RTO/PF4/H complexes. (**F**) KKO/PF4/H complexes stiffened cells on day 1 and softened cells on day 3, but then stiffened cells again on day 6. *n* = 2 independent experiments. Statistics by ANOVA tests were obtained by comparing whole data points (up to 500 points) collected from two independent experiments: ns = no significant difference (*p* < 0.05), *** = significant difference (*p* < 0.05).

**Figure 4 life-11-00832-f004:**
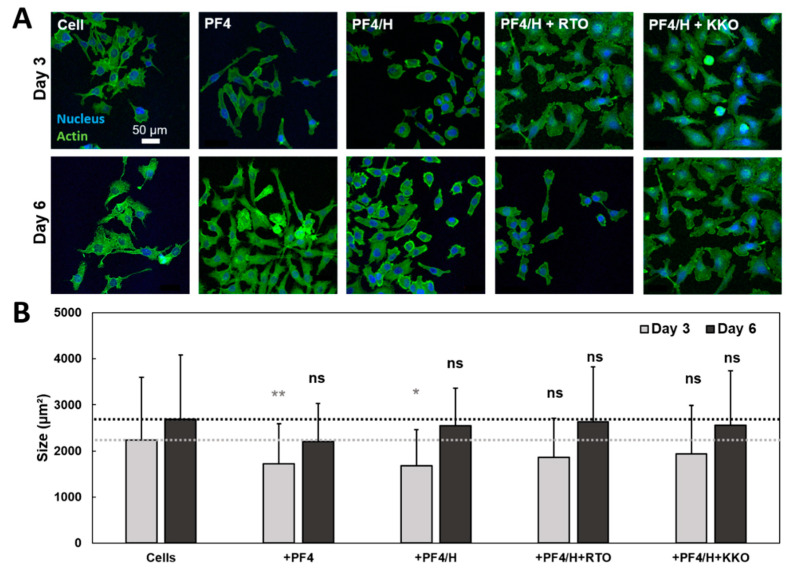
Cell morphological change induced by HIT components. (**A**) Cells were incubated with PF4, PF4/H complexes, and RTO or KKO in complexed with PF4/H and imaged by CLSM on (top panel) day 3 and (lower panel) day 6. Actin was stained by DY590 phalloidin (green) and nucleus by Hoechst 33258 (blue). (**B**) Quantification of cell areas on day 3 (gray) showed a significant reduction in the size of cells caused by PF4and PF4/H, but minimal changes induced by RTO or KKO in complexed with PF4/H. On day 6 (black), the cell’s size increased in all samples. *n* = 2 independent experiments. Statistics by student t-tests were obtained by comparing cell sizes from two independent experiments: ns = no significant difference; * = difference *(p* < 0.05); ** = significant difference (*p* < 0.01) compared to the cell control. Among experiments, the same trend of changes in cell size was observed but the absolute values varied as different cell passages were used, resulting in large standard deviations.

**Figure 5 life-11-00832-f005:**
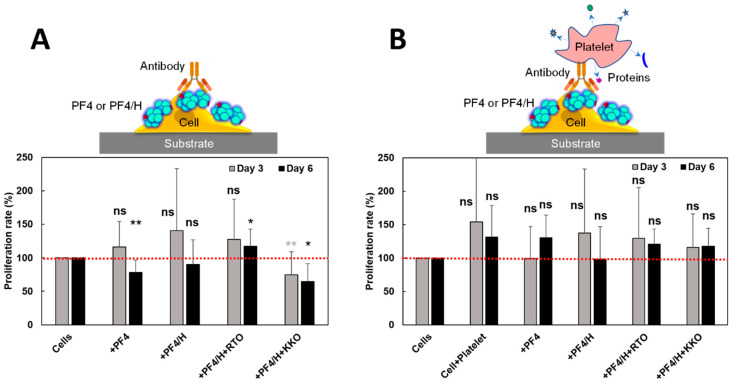
Interference of HIT components and platelets on cancer cell proliferation. (**A**) Cells incubated with HIT components for 3- (gray) and 6 (black) days (*n* = 2 repetitions). (**B**) Cells were incubated with HIT components and platelets for 3- and 6 days (*n* = 3 repetitions). Statistics by t-tests were obtained by comparing cell sizes from two independent experiments: ns = no significant difference (*p* > 0.05); * = difference (*p* < 0.05); ** = significant difference (*p* < 0.01).

## Data Availability

Exclude this statement because the study did not report any data.

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
