# Peer review of "Effect of HIT Components on the Development of Breast Cancer Cells"

_life, 2021, doi:10.3390/life11080832_

Round 1
Reviewer 1 Report
Chen and co-workers have submitted a revised version of their manuscript in which some statistical analysis is present. Unfortunately this analysis, as displayed in the figures, makes clear the inconclusive nature of the experimental data and is in places difficult to believe.
Fig, 2: The authors’ rebuttal states that ‘All E-modulus values from 2 different experiments’. Text at the top of page 7 says ‘all three independent experiments were collected in a histogram distribution’. The caption to Fig. 2A refers to ‘representative histograms of a single experimental series’.
Fig. 3: The new data and statistical analysis here are seriously unconvincing. It is difficult to believe the estimates of significance given the substantial and sometimes complete overlap of the very wide error bars. N=2 is not enough for data of this variability.
Fig. 4B: Again N=2 is not enough. Statistical significance should be calculated between the control and treated cells, not the 3-day and 6-day values for each treatment.
Fig. 5: Very few of these results show any significance, and do not provide strong support for the title and caption of the figure.
In short, the experimental results presented do not provide a sound basis for the Discussion and Conclusions of the paper.
Author Response
Response to Reviewer 1
We thank Reviewer 1 for your very helpful suggestions and comments that helped us to significantly improve our manuscript. Below is our response to your last comments:
Fig, 2: The authors’ rebuttal states that ‘All E-modulus values from 2 different experiments’. Text at the top of page 7 says ‘all three independent experiments were collected in a histogram distribution’. The caption to Fig. 2A refers to ‘representative histograms of a single experimental series’.
Response: We thank the reviewer for pointing out this mistake. We have performed 2 independent experiments. We now changed the text at the top of page 7 to ‘independent experiments’. However, the caption for Fig. 2A was correct. We analyzed each experiment independently and then averaged the values obtained from Gaussian fits for the final values that were presented in Fig. 3.
Fig. 3: The new data and statistical analysis here are seriously unconvincing. It is difficult to believe the estimates of significance given the substantial and sometimes complete overlap of the very wide error bars. N=2 is not enough for data of this variability.
Response: We saw now the problem of unclear description in our data analysis section. The statistics were not obtained by comparison of only 2 average values but they were induced by evaluating all data points from each condition, i.e. up to 500 data points collected from 14 distinct cells. We think that our statistical analysis from 2 independent experiments is sufficient in this case because a high number of data points were presented. Among experiments, we always observed the same trend of changes but the absolute values varied as different cell passages were used. Therefore, the standard deviation presented in Fig. 3 was rather large. We now describe clearly in the figure legends of Fig. 2 and Fig. 3.
Fig. 4B: Again N=2 is not enough. Statistical significance should be calculated between the control and treated cells, not the 3-day and 6-day values for each treatment.
Response: Similar to the indentation experiments, we always observed the same trend of change in the size of cells in these two experiments even though the absolute values slightly differ. Therefore, we performed only n=2 in Fig. 4. We now added this description in the legend of Fig. 4. We now also removed the statistical significance between days 3 and 6.
Fig. 5: Very few of these results show any significance, and do not provide strong support for the title and caption of the figure
Response: We observed in Fig. 5A a clear but different effect on cell proliferation depending on each HIT component, e.g. on day 6, PF4 or PF4/H complexes or KKO reduced proliferation of ~23%, ~11%, or ~36%, respectively, however, RTO/PF4/H complexes enhanced ~17% cell proliferation rate. After adding platelets (Fig. 5B), HIT complexes also enhanced proliferation rate up to ~30% (e.g. by PF4/H or RTO or KKO). Therefore, we believe that the results strongly support the hypothesis that we proposed in the title ‘Effect of HIT components on the development of breast cancer cells’.

Reviewer 2 Report
The uthors improved the manuscript considerably, I would support its publication.
Minor mistake worth to be corrected: The spring constant of cantilevers is measured in N/m not in nm/V (this is optical sensitivity)
Author Response
Response to Reviewer 2
We thank Reviewer 2 for your very helpful suggestions and comments that helped us to significantly improve our manuscript. Below is our response to your last comments:
Minor mistake worth to be corrected: The spring constant of cantilevers is measured in N/m not in nm/V (this is optical sensitivity)
Response: We thank the reviewer for bringing the mistake to our notice. We have now changed the units to N/m.
Round 2
Reviewer 1 Report
None.
This manuscript is a resubmission of an earlier submission. The following is a list of the peer review reports and author responses from that submission.
Round 1
Reviewer 1 Report
The manuscript presents results on the elasticity alterations of metastatic breast cancer cells upon treatment with HIT components. The subject is interesting, the methods are adequate, however in some point more explanations would be required. Furthermore, the low number of parallel experiments hiders the interpretation of the results. I would support the publication of the manuscript after addressing all the questions and comments.
Questions and comments:
- Why was necessary to cover the used spheroidal probes with a hydrophilic (and relatively long) linker?
- The authors state that the spring constant of the used cantilevers was determined prior to each experiment, they present the value of 0.321N/m. Is this nominal value, or the average of all determined values? If so, some deviation would be good to prove the uniformity.
- For each investigated condition 100 force curves were recorded on 10 single cells. 100 on each single cell (which end up in 1000 curves) or 100 curves in total (10 curves each)?
- For the morphological change the fluorescence signal was quantified by ImageJ. How was this quantification effectuated? The reason this is important, that the presenting figure 4 panel B labels the y axis as Size (µl) which is a volume metric. How was the volume extracted from 2D fluorescent images?
- The presented results are claimed to be the average of 3 parallels, which is a low number. However, if all these show the same results, they might be credible. My question is can you present some overall comparison of reproducibility?
- Were the nanoindentation measurements carried out in the same buffer as the cell growth? The high (10%) of serum might influence the concentration of the treatments, as the albumin easily forms complexes with a large range of proteins.
- The very same cultures were measured on the presented days, or simultaneously started parallels were examined on the named days? How is related the uniformness of elastic parameters of cells in one measured petri dish to the found alterations due to treatments?
- The discussion starts with stating increased E modulus occurs due to heparin, which is claimed to be due to depletion of actin filaments. The statement is ment to be supported by REF 42. This sounds controversial, as the depletion of actin filaments would result lower E modulus (as indicated in the shown reference in other cell types).
- The conclusion claims that strong reduction of the cells’ E modulus is induced by the presence of the studied proteins , however the numbers presented for E on the graphs are higher for treatments. Please clarify.
- The authors show that KKO and RTO antibodies have different effects. Some description on their binding mechanism, or differences in their structure which might explain the results would be appreciated and improve the conclusions.
Reviewer 2 Report
Metastasis involves close interactions between cancer cells and circulating platelets. Under certain circumstances, platelet factor 4 (PF4) can aggregate to give an immunogenic complex. This is most commonly triggered by the presence of the anticoagulant polysaccharide heparin, and leads to heparin induced thrombocytopenia (HIT), so the resulting antibodies to PF4 complexes are called HIT antibodies.
In this study the authors investigate the effects of HIT complex components and HIT antibodies on cultured breast cancer cells, in terms of cell elasticity (by atomic force microscopy) cell morphology (by confocal microscopy) and cell proliferation. Some of the results are clear-cut but the statistical significance of others is not made clear (see below). On the whole, this preliminary study established that HIT antibodies may interact with cancer cells and platelets to encourage metastasis. Further study will be needed in the future, for example varying concentrations to obtain dose response curves, and increasing the number of observations so that the results are more robust.
Care is needed to avoid over-interpreting results that do not meet statistical significance.
- Figure 2b and Fig. 3 do not contain any statistical analysis. Values for the number of observations n, and indicators of statistical significance must be added to the figures and captions.
- 4B has some statistical analysis but needs checking. The *** status for comparison between D6 of cells only and D6 of cells+KKO+PF4/H seems surprising.
- 5 also has some statistical analysis but it is noticeable that none of the results are labelled ns for not significant. For example, none of the results in Fig. 5B are labelled – were they not tested for significance?
Non-significant results should not be relied upon to make deductions in the Discussion, though they can be noted as interesting for follow-up in future studies.
Reviewer 3 Report
The manuscript refers to an important problem of the interplay of platelets, PF4, heparin and antibodies and their role in cancer metastasis. I find it descriptive and the interpretation of data found is in many instances rather unjustified. Unfortunately, it contains many errors, even in the description of the data found. Therefore, I recommend to reject it in its present form.
Major remarks
The authors ascribe the changes of the cell elasticity solely to the change of the cytoskeletal structure and depletion of actin filaments and neglect the fact that the elasticity of the cell membrane could have changed also due to its coating with the HIT components, especially with cationic PF4, which is expected to be particularly strongly attracted by the negatively charged cell membrane.
Page 4. The authors first incubated the cancer cells with PF4, H, PF4/H, PF4/H+RTO, PF4/H+KKO and later added platelets. The incubation of the cells with platelets and later adding the HIT components seems to be a better reproduction of the in vivo conditions.
Page 7. The patterns of elasticity change with time for PF4 and PF4/H+KKO are not similar.
Figure 4. The size of cells was measured in microliters?
Page 9, Figure 5.
The discussion of Figure 5 is wrong in many points. On day 6 PF4/H+RTO increased cell proliferation, not decreased.
The data for day 6 are given in black not gray.
The effect for PF4/H was comparable to that of PF4.
The sentence “The PF4 and KKO induced the strongest increase in cell stiffness and strongly inhibited cell proliferation, indicating a clear effect of these proteins on cell growth” is a tautology.
“On day 3, only PF4 inhibited cell proliferation whereas other proteins promoted cell proliferation” – within the experimental error PF4 and some of the proteins/complexes did not change the proliferation rate.
Page 10.
“The changes in cell stiffness have a strong correlation with cell spreading” – for 2 data points one can not claim “a strong correlation”.
“However, in the absence of platelets, PF4 inhibited cell proliferation only on day 3 (Fig.5A)” - on day 3 PF4 increased the proliferation rate, not decreased.
“However, in the presence of platelets, we found that PF4 strongly promoted cell proliferation.” – on day 6 it increased cell proliferation, but on day 3 it inhibited proliferation.
“The multifunctional intervenors in cell communication of heparin lead to poor spreading and thereby stiffen the cells as we observed in our experiments” – the sentence is incomprehensible.
“Most probably, the electrostatic interaction played an important role in the binding of PF4 or heparin to the cells within a short interaction time” – why should binding of PF4 or heparin to the cell surface change with time? There is no reason for this.
“In contrast, heparin which is a highly negative polysaccharide requires a longer time to overcome a surface potential repulsion with the cell membrane before it can interact with cell fibers or interfere with cells” – again, there is no reason why the repulsion between heparin and cell membrane should change with time.
“This stiffness enhancement was not only due to the doubled effect of both PF4 and heparin…” PF4/H is an entity completely different than both of its components therefore one should not expect it will show enhanced effect of the components.
Minor remarks
Page 1: “Under favorable conditions…” – from the point of view of the patient these conditions are very unfavorable.
Page 2, lines 1-2. Heparin is the only polysaccharide used as anticoagulant.
The meaning of the abbreviations KKO and RTO and others (PF4/P, DPBS, PFA) should be explained at first use.
Page 3: The units of the bead diameter are not given.
Language needs to be improved, e.g. “complex interaction” rather than “complex cross-link”. Jargon should be avoided, e.g. “20 ml of cells”.